# Identification and Characterization of Circular RNAs in Mammary Tissue from Holstein Cows at Early Lactation and Non-Lactation

**DOI:** 10.3390/biom12030478

**Published:** 2022-03-21

**Authors:** Yan Liang, Qisong Gao, Haiyang Wang, Mengling Guo, Abdelaziz Adam Idriss Arbab, Mudasir Nazar, Mingxun Li, Zhangping Yang, Niel A. Karrow, Yongjiang Mao

**Affiliations:** 1Key Laboratory for Animal Genetics, Breeding, Reproduction and Molecular Design of Jiangsu Province, Department of Animal Breeding and Production, College of Animal Science and Technology, Yangzhou University, Yangzhou 225009, China; mz120181016@yzu.edu.cn (Y.L.); 18305182715@163.com (Q.G.); hyangwang@163.com (H.W.); gl18852720440@163.com (M.G.); arbabtor@yahoo.com (A.A.I.A.); drmudasirnazar457@gmail.com (M.N.); limingxun@live.com (M.L.); yzp@yzu.edu.cn (Z.Y.); 2Joint International Research Laboratory of Agriculture and Agri-Product Safety of Ministry of Education of China, Yangzhou University, Yangzhou 225009, China; 3Biomedical Research Institute, Darfur University College, Nyala 63313, Sudan; 4Center for Genetic Improvement of Livestock, Department of Animal Biosciences, University of Guelph, Guelph, ON N1G 2W1, Canada; nkarrow@uoguelph.ca

**Keywords:** circular RNA, mammary tissue, Holstein cows, RNA sequencing

## Abstract

In this study, circular RNAs (circRNAs) from Holstein cow mammary tissues were identified and compared between early lactation and non-lactation. After analysis, 10,684 circRNAs were identified, ranging from 48 to 99,406 bp, and the average size was 882 bp. The circRNAs were mainly distributed on chromosomes 1 to 11, and 89.89% of the circRNAs belonged to sense-overlapping circRNA. The exons contained with circRNAs ranged from 1 to 47 and were concentrated from 1 to 5. Compared with the non-lactating cows, 87 circRNAs were significantly differentially expressed in the peak lactation cows. There were 68 upregulated circRNAs and 19 downregulated circRNAs. Enrichment analysis of circRNAs showed that GO analysis mainly focused on immune response, triglyceride transport, T cell receptor signaling pathway, etc. Pathway analysis mainly focused on cytokine-cytokine receptor interaction, T helper 17 cell differentiation, fatty acid biosynthesis, the JAK-STAT signaling pathway, etc. Specific primers were designed for two proximal ends of the circRNA junction sites to allow for PCR validation of four randomly selected circRNAs and carry out circRNA-miRNA interaction research. This study revealed the expression profile and characteristics of circRNAs in mammary tissue from Holstein cows at early lactation and non-lactation, thus providing rich information for the study of circRNA functions and mechanisms, as well as potential candidate miRNA genes for studying lactation in Holstein cows.

## 1. Introduction

Circular RNAs (circRNAs) are a new class of endogenous non-coding RNA molecules formed by covalent bonds that were firstly identified in plant viroids [1]. Compared with traditional linear RNA, circRNAs do not have 5′ and 3′ ends because they are covalently bonded in a circular atresia structure (circRNA is a closed loop, connected at both ends by linear RNA), making them more resistant to RNase R and degradation [2]. Recently, studies have found that circRNAs are closely associated with the occurrence of certain human diseases, which has made circRNA a new trending area of RNA research [3]. CircRNAs are derived from trans-splicing of RNA precursors (pre-RNA) and can be divided into five categories according to their nucleotide source: EciRNA (exonic circular RNA, i.e., all exon-derived circRNA), IciRNA (intronic circular RNA, i.e., all derived from introns), EIciRNA (exon-intron circular RNA, i.e., sense-overlapping circular RNA derived from exons and introns), antisense circular RNA and intergenic circular RNA. CircRNAs from different sources can have different biological functions. Some can adsorb miRNAs, and therefore, regulate the expression of target genes by acting as miRNA sponges [4]. In addition, circRNAs can bind to transcriptional regulatory elements or interact with proteins to regulate gene transcription, and they play a role in m6A (N6-methyladenosine) modification to promote the effective initiation of protein translation [5,6].

Human medical research involving circRNA has mainly focused on cancer, nervous system diseases and immunity [7,8], while livestock circRNA research is still in its infancy. Liang et al. identified 5934 circRNAs from nine different Guizhou miniature pig tissues, including fat and heart and skeletal muscle at three developmental stages, and mapped the temporal and spatial expression profile of these circRNAs to construct the first miniature pig circRNA database [9]. Li et al. analyzed porcine transcriptome data to identify and characterize the function of circRNAs in the heart, liver, spleen, lung and kidney tissue [10]. In terms of lactation research, Zhang et al. and Wang et al. characterized differentially expressed circRNAs in Holstein cattle and sheep mammary tissues at different lactation stages, respectively [11,12].

The mammary gland is an important organ in Holstein cows as it is required for calf survival, passive immunity and early nutrition, and to produce the range of dairy products we consume. The hormones associated with lactation and the growth and apoptosis of mammary epithelial cells show temporal changes during different lactation stages [13], and genetic polymorphisms are known to affect the milk yield and composition [14]. Previous studies suggested that bovine mammary gland development may partly depend on specific microRNA (miRNA) expression patterns [15], and circRNA has been shown to absorb miRNA and thereby contribute to regulating the generation of unsaturated fatty acids by bovine mammary epithelial cells [16,17].

Given the potential of circRNA to indirectly regulate mammary tissue gene expression, it is necessary to identify and characterize circRNAs in mammary tissues at different lactation stages since this circRNA may be involved in the epigenetic and genetic regulation of mammary tissue functions.

This study used high-throughput RNA sequencing (RNA-seq) to study the expression profile of circRNA from Holstein cows during early lactation and non-lactation, and gene ontology (GO) enrichment analysis of the parental miRNA genes of the differentially expressed circRNA was performed. Four differentially expressed circRNAs were selected for authenticity verification by PCR and to investigate circRNA-miRNA interactions.

## 2. Materials and Methods

### 2.1. Statement of Animal Ethics

All experiments were performed in agreement with the care and use guidelines for experimental animals established by the Ministry of Science and Technology of the People’s Republic of China (approval number 2006-398). The mammary gland tissue sample collection process was in line with the welfare ethics of experimental animals, and a production license for experimental animals was obtained (SYDW-2019005). The experimentation was also approved by Yangzhou University, Yangzhou, China.

### 2.2. Animal Sample Collection

In this study, mammary gland tissues were sampled from three Holstein cows in early lactation (*n* = 3, 30 days postpartum) and non-lactation (*n* = 3, 315 days postpartum) on a large dairy farm in Jiangsu province. From the three cows sampled in the study, we collected mammary tissue samples in early lactation and again in non-lactation. All samples were from second-parity cows. Milk was completely extruded from the mammary gland of lactating cows free of mastitis before mammary gland tissue samples were collected. The biopsy sampling methodology is detailed by Li et al. [18]. Briefly, cow hair was shaved from the skin of the sample sites, then the skin was disinfected with ethanol (75%) and locally anesthetized with 1 mL of procaine administered subcutaneously. Next, a 1.5-cm incision was made at the sampling site, and the connective tissues were removed with sterile scissors and forceps to expose the parenchymal tissues. Then, mammary gland tissue biopsies (1–2 g) were harvested, washed with PBS buffer (Invitrogen, Carlsbad, CA, USA) and immediately frozen in liquid nitrogen until RNA was isolated. Finally, after the collection of mammary gland tissue samples, an 11-mm wound clip was used to clamp closed the skin incision, and povidone iodide cream was evenly applied to the skin incision.

### 2.3. RNA Preparation

For RNA extraction, the mammary tissue was treated with TRIzol reagent (Invitrogen, Carlsbad, CA, USA), and the total RNA was then extracted using the RNAprep Pure Tissue Kit (Tiangen, RNAprep Pure Tissue Kit, Beijing, China). Briefly, frozen mammary tissue (10–20 mg) was combined with 300 μL lytic solution RL and was thoroughly ground with a grinding pestle. Then, 590 μL RNase-free ddH2O and 10 μL Proteinase K were added to the homogenate and mixed at 56 °C for 10–20 min. After dissolving the total RNA with appropriate diethylpyrocarbonate (DEPC), the quantity and quality of RNA were measured using a spectrophotometer (NanoDrop^®^ ND-1000, Thermo Scientific, DE, Waltham, MA, USA). The quantity of total RNA was greater than 400 ng μL^−1^, and the 260/280 requirement was 1.9~2.0.

### 2.4. Identification of circRNAs in Mammary Tissue from Holstein Cows

The CircBase database included circRNA sequences of five species: human, mouse, nematode, cactus and coelacanth. If a species to be analyzed belongs to one of the above species, the circRNA identified is first studied using CIRI software (Gao, CircRNA Identifier, 2015) [19], and the identified results are compared with the above database to obtain the known circRNA and the newly identified circRNA. If the species does not belong to the above species, circRNA is predicted ab initio using CIRI. In our sequencing, Holstein cows did not belong to the database and thus needed to be predicted from scratch.

A sequencing library was constructed, and genomic sequence mapping and analysis were performed. Ribosomal RNA was removed from the mammary tissue RNA samples using a transcriptome isolation kit (Ribominus Bacteria 2.0, Thermo Fisher). The remaining RNA were paired-end sequenced using an Illumina HiSeq Xten (Illumina Inc., San Diego, CA, USA) from Shanghai Personal Biotechnology Company, Ltd. (Shanghai, China). After sequencing, the data were preprocessed as follows: (a) extreme reads of signal strength caused by sequencing instrument hardware were removed; (b) reads with low overall quality (Q = 20, base proportion less than 50%) were removed; (c) the proportion of read bases with an error rate less than 1% was removed; (d) reads with N-base ambiguity caused by insufficient sequencing fluorescence intensity were removed; (e) reads with a length of fewer than 20 bases and containing adaptor sequences were removed; (f) ribosomal RNA reads were removed. The clean reads obtained through the above preprocessing were used to identify the circRNAs using the website find_circ (https://github.com/marvin-jens/find_circ, accessed on 25 April 2021) [20]. The known circRNAs and newly predicted circRNAs were obtained using the CIRI software to predict the circular RNAs, and by comparing data with the circBase database (http://circrna.org/cgi-bin/singlerecord.cgi?id=mmu_circ_0001771, accessed on 26 April 2021) [19]. The chromosome distribution and length distribution of the identified circRNAs were analyzed according to the find_circ search Deseq website (http://bioconductor.org/biocLite.R, accessed on 28 April 2021), which was used to conduct standardized processing of the number of junction read counts of circRNA in each sample (base mean value was used to estimate the expression level). The difference multiple was also calculated, and NB (negative binomial distribution test) was used to test for a different significance of read numbers [21]. Finally, the differentially expressed circRNAs were screened according to the difference multiple and difference significance test results (Appendix B, Table A1).

### 2.5. PCR Validation of circRNAs

Specific primers were designed for both ends of the circRNA junction site for reverse transcriptase-polymerase chain reaction (RT-PCR) amplification, to verify the existence of circRNA_09759, circRNA_09048, circRNA_09761 and circRNA_03309 (Table 1). The RT products were checked for size and purity by 1% agarose gel electrophoresis (voltage 100 V, 30 min) and then sequenced by the Shanghai Sangon Company (Shanghai, China). Three software programs, SeqMan (Invitrogen, Carlsbad, CA, USA), SnapGene Viewer (Invitrogen, Carlsbad, CA, USA) and Vector NTI(Invitrogen, Carlsbad, CA, USA) were then used to analyze the sequences and find the junction sites in the circRNAs.

### 2.6. Target microRNA Predictions and Gene Ontology Enrichment Analysis

The miRNA-targets of each differentially expressed circRNA were predicted using the miRanda algorithm [22], and the interaction network of the circRNAs and their target miRNAs was analyzed using starBase and then drawn using Cytoscape [23]. The calculation method of ceRNA_score and *p*-value [24] is as follows:(1)$$\mathrm{ceRNA}\_\mathrm{score}\;=\frac{\#\mathrm{MRE}\_\mathrm{for}\_\mathrm{share}\_\mathrm{miRNA}}{\#\mathrm{MRE}\_\mathrm{for}\_\mathrm{circRNA}\_\mathrm{miRNA}}$$
where circRNA represents the information of circRNA; ceRNA_score represents predicted ceRNA relationship scores; #shared_miRNA represents the number of co-miRNAs; miRNAs represent the names of co-miRNAs; *p*-value represents the *p*-value for ceRNA prediction.

The calculation formula of the *p*-value is as follows:(2)$$p=\overset{\min\left(m_{p},m_{n}\right)}{\underset{i=m_{c}}{\sum }}\frac{\left(\begin{array}{c} m_{n}\\ i \end{array}\right)\left(\begin{array}{c} M_{T}-m_{n}\\ m_{p}-i \end{array}\right)}{\left(\begin{array}{c} M_{T}\\ m_{p} \end{array}\right)}$$
where *M_T_* represents the number of all miRNAs; *m_p_* represents the number of miRNAs that regulate this mRNA; *m_n_* represents the number of miRNAs that have regulatory effects on the circRNA; *m_c_* represents the number of common miRNAs.

Gene Ontology (GO) enrichment analysis was used to investigate the main functions of the parent genes of the differentially expressed circRNAs using the DAVID tool (https://david.ncifcrf.gov, accessed on 12 June 2021) [25,26].

## 3. Results

### 3.1. Identification and Sequence Characteristics of circRNAs in Mammary Tissue from Holstein Cows

A total of 10,684 circRNAs (Appendix C, Table A2) were identified from RNA in mammary tissue from Holstein cows by library construction, sequencing and bioinformatics analysis (detected in the actual sequencing were 10,684 GT-AG Splicing_signals). In total, 3250, 2768 and 3098 circRNAs were predicted in Holstein cows’ mammary tissue at 30 days, and 4143, 3765 and 3359 circRNAs were predicted at 315 days of lactation, respectively (Figure 1). The identified circRNAs were mainly distributed on chromosomes 1 to 11 (Figure 2A). Chromosome 1 contained the most circRNAs (*n* = 629). The sizes of circRNAs ranged from 48 to 99,406 bp, and the average size was 882 bp. The circRNA lengths mainly ranged by 201–400 bp, though some were greater than 2000 bp (Figure 2B). The number of exons contained with circular RNAs ranged from 1 to 47 and these were concentrated between 1 and 5 (Figure 2C). Variable shear signals’ AT reverse shear sites in circRNA sequences were counted, and all of them were CG-AT (Figure 2D). The CG content of circRNA was distributed in the range of 30–80%, mainly concentrated in the range of 40–50% (Figure 2E). It was found that 89.89% of circRNAs belong to EIciRNA, 4.58% belong to IciRN, and only 3.22% belong to the EciRNA (Figure 2F).

### 3.2. Differential Expression of circRNAs: Analysis in Mammary Tissue from Holstein Cows at Early Lactation and Non-Lactation

Compared with the nonlactating Holsteins, 87 circRNAs with significantly different expressions were identified in the early lactation group, 68 of which were upregulated and 19 were downregulated (Table A1 and Figure 3, specific data are presented in the Appendix A). To describe the functions of differentially expressed circRNAs, GO enrichment analysis was carried out on their target genes. The GO analysis identified immune response, triglyceride transport and T cell receptor signaling functions (Figure 4). KEGG pathway analysis mainly focused on cytokine-cytokine receptor interactions, Th17 cell differentiation, fatty acid biosynthesis and the JAK-STAT signaling pathway (Figure 5).

### 3.3. CircRNA Authenticity Verification

CircRNA_09759, circRNA_09048, circRNA_09761 and circRNA_03309 were selected from the differentially expressed circRNAs for authenticity verification. These circRNAs are from mitochondrial glycerol-3-phosphate acyltransferase (GPAM), tumor necrosis factor receptor superfamily-member 21 (TNFRSF21) and solute carrier family 27 isoform A6 (SLC27A6), and may play a role in fatty acid transport, triglyceride synthesis, inflammation and immune regulation (Appendix A). PCR amplification validated the existence of the circRNAs (Figure 6), and they produced the expected band sizes on the agarose gel (215 bp, 202 bp, 173 bp and 221 bp, respectively) (Figure 6A). DNA sequencing confirmed the presence of head-to-tail splice junctions as suggested by the RNA-seq analyses (Figure 6B) and the size of the circRNAs.

### 3.4. CircRNA-miRNA Interaction Research

The predicted ceRNA relationships were paired and sorted from largest to the smallest according to the ceRNA_score and shared miRNA. Records with shared miRNAs less than 3 and *p*-values greater than 0.05 were filtered out [27]. The filtered results were shown in Figure A1 (Appendix D). We selected the four differentially expressed circRNAs from Table A1 (Appendix B) and the results showed that circRNA_08052 had regulatory relationships with AC_000178.1_22528, AC_000169.1_14270, bta-miR-154b and bta-miR-451; circRNA_03706 had regulatory relationships with bta-miR-6522, bta-miR-2411-5p, AC_000159.1_2251 and bta-miR-1298; circRNA_02728 had regulatory relationships with bta-miR-154b, bta-miR-182, AC_000160.1_3579, bta-miR-223, bta-miR-370, bta-miR-493 and bta-miR-146b; circRNA_09048 had regulatory relationships with bta-miR-493, bta-miR-370 and bta-miR-154b (Figure 7).

## 4. Discussion

Cows are not only a major source of dairy products but also an ideal large animal model to study the transcriptome and expression characteristics of the mammary gland. As a new non-coding RNA, circRNA has become a new research hotspot in recent years. Many studies have reported that circRNA is widely present in humans [28], mice [29], pigs [30], cattle [31], sheep [12] and other species [32]. 

Studies have identified circRNAs in sheep breast tissues during early lactation and non-lactation and found 3278 and 1756 circRNAs, 40 of which were upregulated and one of which was downregulated [12]. Hao [33] identified 4906 circRNAs in two sheep mammary gland tissues with different lactation performance levels, and 33 of these were differentially expressed between breeds. Another study showed that 6621 circRNAs were differentially expressed in the mammary tissue of Holstein cows at 90 and 250 days postpartum, of which 2231 were co-expressed [11]. In this study, high-throughput sequencing was used for the first time to explore the presence and expression of circRNAs in mammary tissue from Holstein cows at peak lactation and during the nonlactating period, and to screen and identify circRNAs that may play an important role in lactation. Through systematic identification and analysis of circRNAs, it was found that 3250 and 3359 circRNAs were predicted in the mammary tissue of Holstein cows at 30 and 315 days postpartum. These different results may be due to the relationship between mammalian mammary tissue development and the hormone level, genetic performance, parity, nutritional status and feeding management. Studies have shown that hormones in bovine mammary epithelial cells under different conditions have certain effects on the expression of genes related to milk composition synthesis [34]. For example, milk-derived hormones affect functional differentiation and local environmental signaling of mammary epithelial cells [35], thereby affecting mRNA expression. Therefore, different species, nutrient levels, physiological stages and external conditions may affect the expression of circRNAs in mammary tissue. 

Most of the circRNAs detected in the mammary tissue were very short, less than 1 kb in length, while some circRNAs were greater than 2 kb in length, which was also consistent with the results of analysis of circRNAs in the mammary glands of cows [11] and sheep [12], as well as bull testicles [36]. In addition, this study found that 89.89% of the identified circRNAs belonged to EIciRNA and were mostly distributed on chromosome 1. Wang et al. [12], meanwhile, identified six types of circRNA in sheep mammary tissue, among which EciRNA was the main one, with the circRNAs mainly concentrated on chromosomes 1 and 3. It is not surprising that bovine chromosome 1 produces the most circRNAs because it is the largest [37].

GO and KEGG enrichment analysis can illustrate the related functions of genes. In this study, the GO entries enriched by differential circRNAs were mainly involved in regulating various signaling pathways, metabolic processes and cell element composition, as well as signal transduction. The GO terms of circRNA parent genes have different enrichment levels at different stages of lactation [11]. The GO entry of immune response is the most significantly enriched biological process, which is important for fighting pathogen infections, for example, mastitis-causing pathogens including *Escherichia coli* and *Staphylococcus aureus* [38,39]. The next enriched biological processes were triglyceride transport and response to fatty acid (FA). In addition, the KEGG pathway analysis mainly revealed cytokine-cytokine receptor interactions, T helper cell 17 (Th17) differentiation-related pathways and biosynthesis of FA, and significantly enriched immune process pathways. Th17a T helper cells are derived from T helper cell 0 (Th0) when stimulated with IL6 and IL23, and are a potent source of inflammatory IL-17, playing an important role in autoimmunity in humans. 

Studies have shown that the CD molecule is often used as an important receptor or ligand for cells [40]. Both CD358 and CD4 are leukocyte differentiation antigens. CD358 is member 21 of the tumor necrosis factor receptor superfamily [41]. Overexpression of CD358 enables its cytoplasmic death domain to induce apoptosis and triggers the NF-κB pathways [42]. As an important member of the FA biosynthesis process, the main role of GPAM is to promote the production of triglyceride (TAG) in animals by catalyzing the biosynthesis of triacylglycerol and phospholipids. SLC27A6 has been characterized as a membrane-associated FA-binding protein that participates in FA transport across a cell membrane and is expressed primarily in heart muscle tissue [43]. Based on this, we selected circRNAs related to CD358, GPAM and SLC27A6 from the differentially expressed circRNAs for authenticity verification, and the existence of four circRNAs (circRNA_09759, circRNA_09048, circRNA_09761 and circRNA_03309) was successfully proven. 

In addition, it has been reported that circRNAs derived from four casein coding genes (CSN1S1, CSN1S2, CSN2 and CSN3) are highly expressed in the mammary tissue of cows after 90 days of lactation, and these circRNAs also have miR-2284 binding sites, which may be involved in the regulation of casein expression [11]. Liang et al. [9] identified 149 circRNAs related to the cation equilibrium, ATP hydrolyzation-coupled cation transport, tight cell junctions and calcium signaling pathways. Sun et al. [32] found that 40 circRNAs were involved in the miRNA-mediated ceRNA regulation network in Lantang pigs. Li et al. [33] found that cirCFGFR4 can absorb miR-107 and promote the expression of the Wnt3A gene, thereby promoting the differentiation of myoblasts and inducing apoptosis. Another study [44] also reported that circ11103 interacts with miR-128 to regulate milk fat metabolism in dairy cows. The above studies found that in different organisms and different physiological stages, circRNA has a specific expression-regulation mechanism and rich and important functions, not just appearing as a by-product of transcription. We also proved this in the circRNA-miRNA interaction research. Looking ahead, we will focus on circRNA_09048 for further study since we found that circRNA_09048 derives from the immune-related gene CD358 and has a regulatory relationship with miR-370 in circRNA-miRNA interaction research.

## 5. Conclusions

In this study, through high-throughput sequencing of circRNAs in Holstein cow mammary tissues at early lactation and non-lactation, 10,684 circRNAs were detected,. These were mainly distributed on chromosomes 1 to 10, with an average size of 882 bp and were mainly EIciRNA. Among the 87 differentially expressed circRNAs detected, enrichment analysis found that they were mainly concentrated on the immune response, T cell receptor signaling pathway, cytokine-cytokine receptor interaction and other pathways. This study revealed the expression profile and characteristics of circRNAs in the mammary tissue of Holstein cows during the early lactation and non-lactation periods and provided a wealth of information for studying the function and mechanism of circRNA.

## Figures and Tables

**Figure 1 biomolecules-12-00478-f001:**
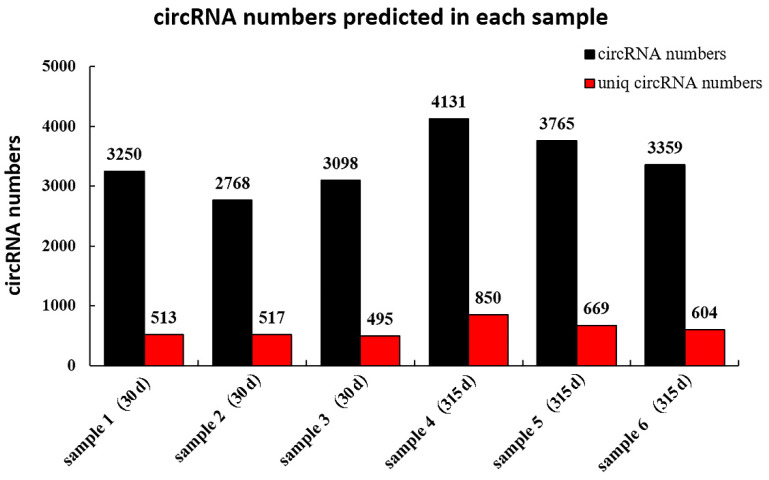
CircRNA numbers predicted in each sample. The vertical axis is the number of circRNAs; the horizontal axis shows the individual samples from cows on postpartum days 30 or 315; the numbers above each bar refer to the number of circRNAs predicted in each sample; the Uniq_circRNA_numbers refer to the number of circRNAs specifically predicted in each sample compared to other samples in the project.

**Figure 2 biomolecules-12-00478-f002:**
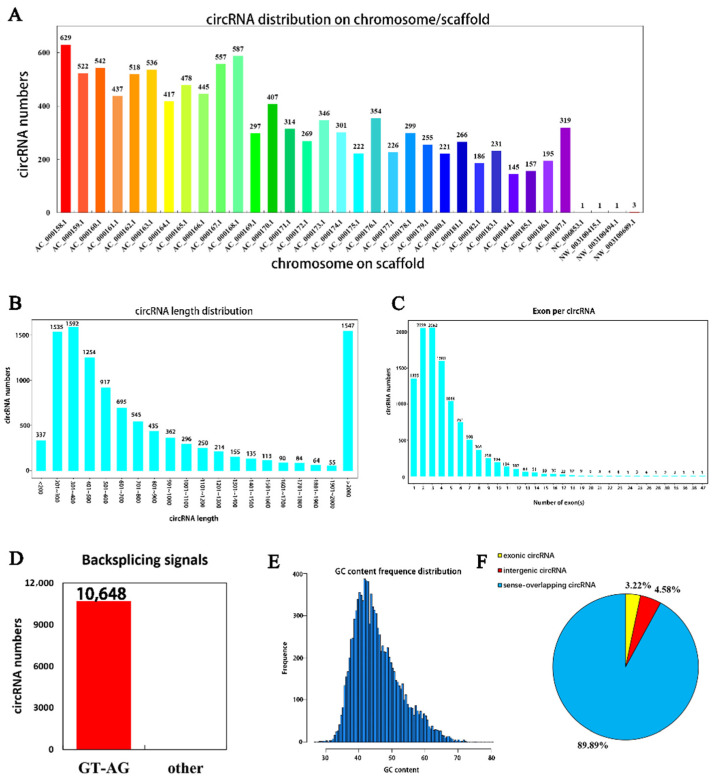
Identification, characterization and chromosomal distribution of circRNAs. (**A**) The number of circRNAs per chromosome; (**B**) the length distribution density of circRNAs); (**C**) the number of exons by circRNAs; (**D**) statistical diagram of circRNA shear signal, with the number of circRNAs on the vertical axis and the type of shear signal on the horizontal axis (specific data are presented in the Appendix A); (**E**) the vertical axis is the number of circRNAs and the horizontal axis shows the CG content of circRNAs; (**F**) percentage of different types of circRNAs.

**Figure 3 biomolecules-12-00478-f003:**
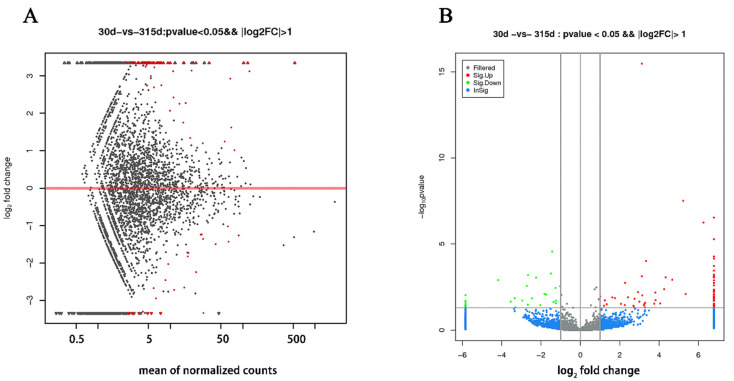
Differentially expressed circRNAs in mammary tissue from Holstein cows at early lactation and non-lactation. (**A**) The *X*-axis is the mean expression of all samples used for comparison after standardization. The *Y*-axis is the Log_2_ fold change. The red highlights are significantly differently expressed circRNAs. (**B**) Gray and blue circRNAs with non-significant differences; red and green circRNAs with upregulated and downregulated significant differences, respectively. The *X*-axis is the log_2_ fold change and the *Y*-axis is the log_10_
*p*-value.

**Figure 4 biomolecules-12-00478-f004:**
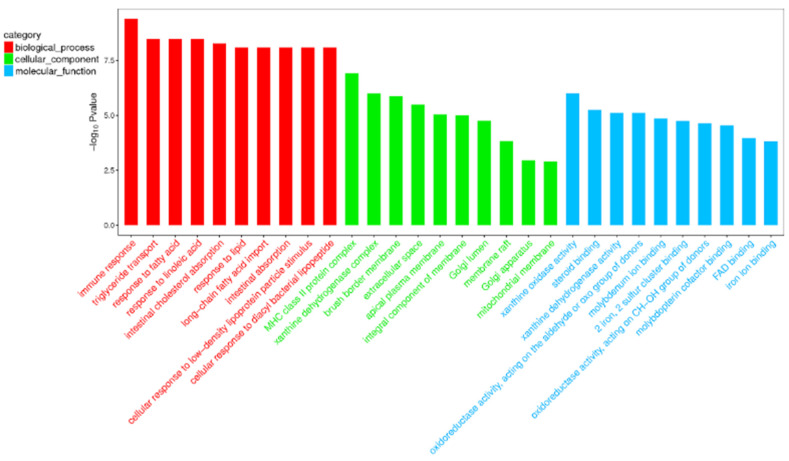
Top 30 categories of GO analysis. The basic information for each node, i.e., the GO ID and GO term, is displayed in the corresponding graph. The *X*-axis is the GO entry name and the *Y*-axis is the log_10_
*p*-value.

**Figure 5 biomolecules-12-00478-f005:**
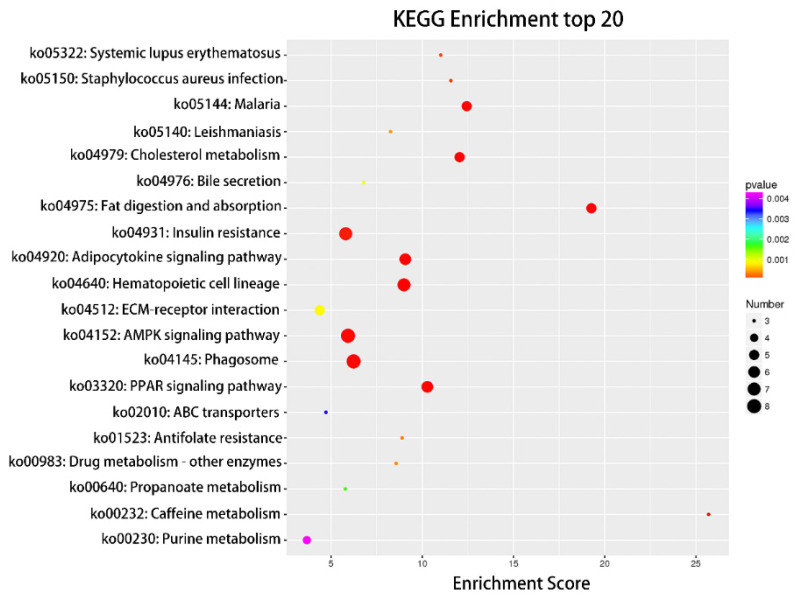
Top 20 categories of KEGG pathway analysis. KEGG enrichment top 20 bubble diagram, where the *X*-axis is the Enrichment Score. The larger the bubble, the more circRNAs the item contains, and the color of the bubble changes from purple to blue to green to red. The smaller the Enrichment *p*-value, the greater the significance.

**Figure 6 biomolecules-12-00478-f006:**
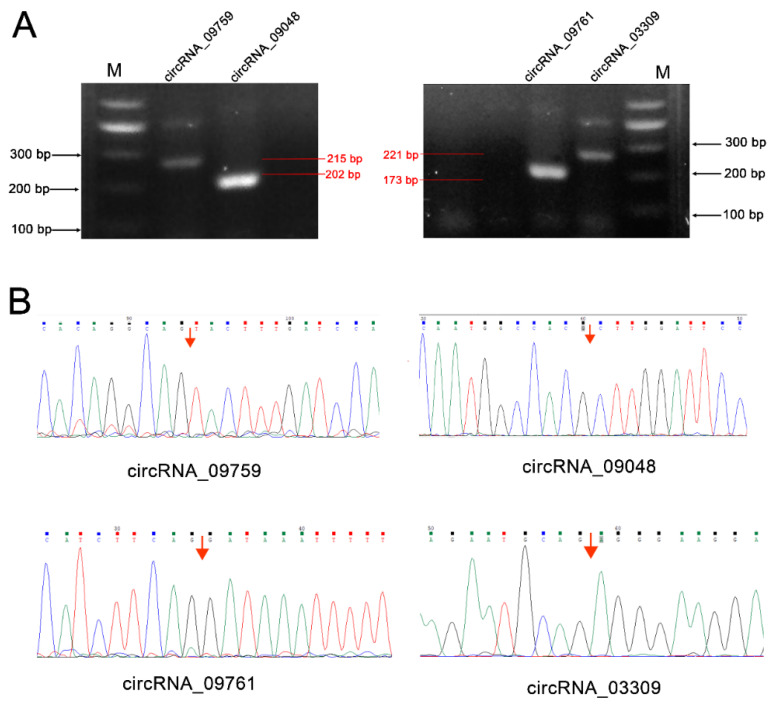
RT-PCR validation of the presence of circRNAs in cow mammary gland tissue. (**A**) Reverse transcriptase-polymerase chain reaction (RT-PCR) amplimers derived from the circular RNAs using divergent primers for cows’ mammary gland RNA (M: marker); (**B**) Head-to-tail splice junctions for the circRNAs were confirmed by DNA sequencing and are marked with a red arrow on the DNA sequence chromatograms.

**Figure 7 biomolecules-12-00478-f007:**
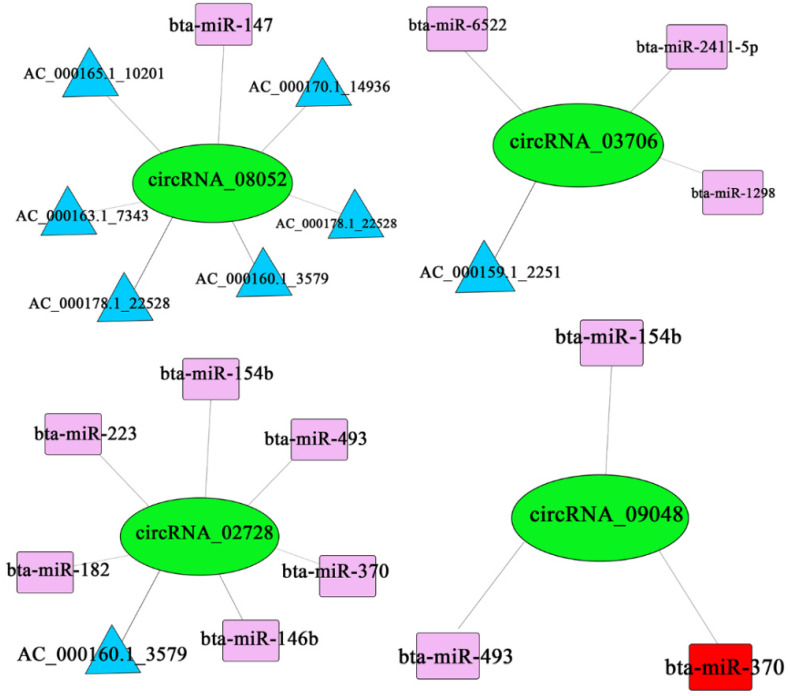
Four-network regulation of mRNA–circRNA–miRNA.

**Table 1 biomolecules-12-00478-t001:** PCR primers used to amplify specific circRNAs.

CircRNA	Forward (5′→3′)	Reverse (5′→3′)	Amplicon Size (bp)
circRNA_09759	TCTGGCTGAGCACATTCTCTTCAC	TCGCAGTAGTTCAACTATATGCCC	215
circRNA_09048	CAGAACCGGGAGAAGTGGATCTAC	ACGCAGGCTTATGTTGGTACAGTG	202
circRNA_09761	TTCAGGCAATAATCTCAACATCCC	TTATGGACATCTCGCTCTTGAATG	173
circRNA_03309	AACCAAACATGTCTTTAGACTTGG	TCCTTCAGTAGCTCCATAAAACTC	221

## Data Availability

Not applicable.

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
