# Peer review of "Identification and Characterization of Circular RNAs in Mammary Tissue from Holstein Cows at Early Lactation and Non-Lactation"

_biomolecules, 2022, doi:10.3390/biom12030478_

Round 1
Reviewer 1 Report
This is an interesting report on the circular RNA expression in the mammary tissue from Holstein cows during the early lactation and dry periods. Using high-throughput sequencing of circRNAs in Holstein cow mammary tissues, the authors identified 10684 circRNAs with the determination of the location of the chromosomes. The study provides important genetic information on genetics and lactation in Holstein cows. However, before it can be published, some clarification and improvement are required for this manuscript.
1. Are those three cows in early lactation and the three non-lactation cows in the same group (total three or total 6 cows)?
2. What is the difference between “identified” and “predicted”? Why did the authors not annotate the “identified” but “predicted”?
3. Does the number 10684 come from the real sequencing data? Is this number the sum of all circular RNA from the sequencing data of 6 samples? Or it is the average of the 6 samples? If this is the sum, then, how many of them are the overlapping circRNA?
4. Figure 1 shows the six numbers (3250, 2768, 3098, 4131, 3765, 3359) of the predicted circRNA from six samples, but the text says that “a total of 3250 and 3359 circRNA were predicted”, why these two numbers were picked?
5. The overall writing is good, there are some minor grammar errors (such as line 29: …4 random selected…” should be “ randomly selected..” Please carefully proofread the manuscript.
Author Response
We will be happy to edit the text further, based on helpful comments from the comments. All of my original comments have been addressed and have minor comments on the revised manuscript:
- Are those three cows in early lactation and the three non-lactation cows in the same group (total three or total 6 cows)?
Response: Thanks for your comment. In our manuscript, three cows in early lactation and the three non-lactation cows in the same group, and the total number of cows is three. We collected mammary tissue samples from the three cows in early lactation (DIM 30) and again in non-lactation (DIM 315). All samples were from 2nd parity cows.
- What is the difference between “identified” and “predicted”? Why did the authors not annotate the “identified” but “predicted”?
Response: Thanks for your comment. CircBase database included circRNA sequences of five species: human, mouse, nematode, cactus and coelacanth. If the species to be analyzed belongs to one of the above species, the circRNA identified is first performed using CIRI software, and the identified results are compared with the above database to obtain the known circRNA and the newly identified circRNA. If the species does not belong to the above species, circRNA is predicted ab initio using CIRI software. In our sequencing, Holstein cows were does not belong to the five species mentioned above and needed to be predicted from scratch, so we chose to use "prediction" rather than "identification".
- Does the number 10684 come from the real sequencing data? Is this number the sum of all circular RNA from the sequencing data of 6 samples? Or it is the average of the 6 samples? If this is the sum, then, how many of them are the overlapping circRNA?
Response: Thanks for your reminder very much.
1.The number 10684 come from the real sequencing data, detected in the actual sequencing were 10684 Splicing_signal (GT; AG).
- These 10684 circRNAs are not simply the sum or average of the six samples. First, different samples from each group also had their uniq circRNAs, it's actually the sum of all circRNAs after pairwise comparison from six samples, excluding the overlapping circRNAs. Second, the comparison between the two groups at different stages, selecting the union of circRNAs in the two groups (early lactation and the non-lactation cows) of samples. The distribution is shown in the table below.
- The number of overlapping circRNAs between the two groups is 1654.
Table 1. The number of circRNAs between the two groups
|
Group |
Sample |
circRNA numbers |
uniq circRNA numbers |
Pairwise comparison in each group |
The overlapped in pairs in each group |
The total number of uniq circRNAs |
The total number of overlapping in each group |
The total numberof circRNAs for each group |
|
Early lactation |
sample 1 |
3250 |
513 |
sample 1 VS sample 2 |
2196 |
1525 |
3764 |
5289 |
|
sample 2 |
2768 |
517 |
sample 1 VS sample 3 |
2679 |
||||
|
sample 3 |
3098 |
495 |
sample 2 VS sample 3 |
2203 |
||||
|
sample 1 VS sample 2 VS sample 3 |
1657 |
|||||||
|
|
||||||||
|
Non-lactation |
sample 4 |
4131 |
850 |
sample 4 VS sample 5 |
2765 |
2123 |
4926 |
7049 |
|
sample 5 |
3765 |
669 |
sample 4 VS sample 6 |
3018 |
||||
|
sample 6 |
3559 |
604 |
sample 5 VS sample 6 |
2931 |
||||
|
sample 4 VS sample 5 VS sample 6 |
1894 |
|||||||
|
|
||||||||
|
The total number of circRNAs between the two groups |
12338 |
|||||||
|
The overlapped between the two groups |
1654 |
|||||||
|
The total number of circRNAs |
10684 |
|||||||
- Figure 1 shows the six numbers (3250, 2768, 3098, 4131, 3765, 3359) of the predicted circRNA from six samples, but the text says that “a total of 3250 and 3359 circRNA were predicted”, why these two numbers were picked?
Response: Thanks for your reminder very much. We apologize for the negligence of our manuscript. We worked on the manuscript for a long time and the repeated addition and removal of sentences obviously led to some problems in the description. We have revised the specific content in the manuscript, as is shown in line 171-173.
The revised content is as follows: “A total of 3250, 2768 and 3098 circRNAs were predicted in Holstein cows' mammary gland tissue at 30 days, and 4143, 3765 and 3359 circRNAs at 315 days lactation, respectively (Figure 1)”.
- Moderate English changes required.
Response: Thank you very much, we have now worked on both language and readability and have also involved native English speakers for language corrections. We really hope that the flow and language level have been substantially improved.

Reviewer 2 Report
In this manuscript, Liang and colleagues analysed the repertoire of circular RNAs (circRNAs) in the mammary tissue from Holstein cows at early lactation.
Mammary tissues samples were taken from cows at early lactation and non-lactating, the RNA was prepared according to standard procedures, and the output from NGS sequencing was analysed the software find_circ and the circBase database. Some differentially expressed circRNas were validated by RT-PCR. This analysis resulted in the identification of 10.684 circRNAs, of which 87 were differentially expressed in lactating versus not-lactating animals. The authors also provide a Gene Ontology (GO) analysis, a KEGG pathways analyses and a circRNA-miRNA interaction prediction analysis to provide some insights into the function and mechanism of the identified circRNA.
Overall, the research presented here shows some points of interest for the broad readership of Biomolecules. The experimental setup looks adequate, and I think the author’s conclusions are supported by their results.
Minor points:
Line 40: “circular atresia structure”: is not clear to me the meaning of “atresia” in this context
Line 106: TRIzol should be instead of TRNzol
Line 118: it should be “ribosomal RNA”
Line 119: why the authors used the Ribominus Bacteria 2.0, Thermo Fisher to deplete ribosomal RNA?
Author Response
We will be happy to edit the text further, based on helpful comments from the comments. All of my original comments have been addressed and have minor comments on the revised manuscript:
- Line 40: “circular atresia structure”: is not clear to me the meaning of “atresia” in this context.
Response: We thank the reviewer for pointing out this issue. In our manuscript, the meaning of “atresia” in this context refers to the fact that the circRNA is a closed loop, connected at both ends by linear RNA.
- Line 106: TRIzol should be instead of TRNzol.
Response: Thanks for your comment, we apologize for the negligence of our manuscript. We mistyped here and we have corrected the word from “TRNzol” to “TRIzol”, as is shown in line 106, thanks again.
- Line 118: it should be “ribosomal RNA”.
Response: Thanks for your reminder very much. We apologize for the negligence of our manuscript. We worked on the manuscript for a long time and the repeated addition and removal of sentences obviously led to some problems in the description. We have revised the specific content in the manuscript, we have corrected the expression from “Ribosomal” to “Ribosomal RNA”, as is shown in line 118.
- Line 119: why the authors used the Ribominus Bacteria 2.0, Thermo Fisher to deplete ribosomal RNA?
Response: Thanks for your comment. As to why we used the kit (Ribominus Bacteria 2.0, Thermo Fisher) to remove ribosomal RNA, there are several reasons:
- Most DNA can be transcribed into RNA, but only 1.5% of nucleotide sequences are used to encode proteins. The remaining non-coding Rnas (Ncrnas) that do not encode proteins are considered genomic transcription noise. This transcriptome noise (invalid information) comes almost entirely from the most abundant member, rRNA.
- The purpose of sequencing is to obtain more biological information, but rRNA, the most abundant member of RNA, can only provide very little information about the transcript, and the detection of too much rRNA will mask the expression richness of other genes. Therefore, rRNA is usually removed from RNA samples prior to sequencing, and the efficiency of rRNA removal is also considered a key factor in maximizing the read of transcripts.
